# Gradual Analytics of Starch-Interacting Proteins Revealed the Involvement of Starch-Phosphorylating Enzymes during Synthesis of Storage Starch in Potato (*Solanum tuberosum* L.) Tubers

**DOI:** 10.3390/molecules28176219

**Published:** 2023-08-24

**Authors:** Junio Flores Castellanos, Arsalan Khan, Joerg Fettke

**Affiliations:** Biopolymer Analytics, Institute of Biochemistry and Biology, University of Potsdam, Karl-Liebknecht-Str. 24-25, Building 20, 14476 Potsdam-Golm, Germany; junio.flores.castellanos@uni-potsdam.de (J.F.C.); arsalan@uni-potsdam.de (A.K.)

**Keywords:** starch, starch-interacting proteins, glucan water dikinase, phosphoglucan water dikinase, plastidial phosphorylase, potato, *Solanum tuberosum* L.

## Abstract

The complete mechanism behind starch regulation has not been fully characterized. However, significant progress can be achieved through proteomic approaches. In this work, we aimed to characterize the starch-interacting proteins in potato (*Solanum tuberosum* L. cv. Desiree) tubers under variable circumstances. Starch-interacting proteins were extracted from developing tubers of wild type and transgenic lines containing antisense inhibition of glucan phosphorylases. Further, proteins were separated by SDS-PAGE and characterized through mass spectrometry. Additionally, starch-interacting proteins were analyzed in potato tubers stored at different temperatures. Most of the proteins strongly interacting with the potato starch granules corresponded to proteins involved in starch metabolism. GWD and PWD, two dikinases associated with starch degradation, were consistently found bound to the starch granules. This indicates that their activity is not only restricted to degradation but is also essential during storage starch synthesis. We confirmed the presence of protease inhibitors interacting with the potato starch surface as previously revealed by other authors. Starch interacting protein profiles of transgenic tubers appeared differently from wild type when tubers were stored under different temperatures, indicating a differential expression in response to changing environmental conditions.

## 1. Introduction

Starch is a carbohydrate polymer synthesized by plants and most algae for energy storage. It is naturally produced as a highly ordered and dense packaging of glucan chains, leading to the formation of insoluble granules located mostly in plastids (chloroplasts and amyloplasts) [1]. The mechanism for synthesis of this stunning structure is complex and involves the synchronized activity of several groups of enzymes just for generating the essential glycosidic bonds. From a general perspective, the major groups of enzymes involved in starch synthesis, each with various specific isoforms, can be classified in starch synthases (SSs, EC 2.4.1.21), starch branching enzymes (SBEs, EC 2.4.1.18), and starch debranching enzymes (DBEs, 3.2.1.41) [2,3,4]. Aside from these core groups of proteins, other enzymes with essential activities in starch biosynthesis take part. ADP-Glc pyrophosphorylase (AGPase, EC 2.7.7.27) is, for instance, the key enzyme on the first steps of starch synthesis, which produces ADP-Glc from ATP and glucose-1-phosphate. ADP-Glc is the substrate used by the SSs to extend the α-1,4 linked glucans. There are five main isoforms of SSs essentially found in all crops, namely SS1, SS2, SS3, SS4, and Granule Bound Starch Synthase (GBSS or WAXY). Two additional isoforms, SS5 and SS6, were recently found to interact with the starch granules of potato tuber [5]. Among these, SS1, SS2, and SS3 are involved in the synthesis of amylopectin, a polymer made of α-1,4 linked glucan chains connected by α-1,6 linkages [6,7]. On the other side, GBSS is involved in the synthesis of amylose, a linear α-1,4 linked glucose polymer. Recently, a protein named Protein Targeting to Starch 1 (PTST1) was discovered to address GBSS to the starch granules [8]. SBEs, on the other side, are responsible for the trimming of α-1,4 glucan chains transferring segments to an acceptor chain to introduce branching points through α-1,6 linkages, having a major role in the formation of the amylopectin structure [9]. DBEs, in counterpart, hydrolyze the α-1,6 glycosidic bonds releasing glucan chains. In this way, DBE also contributes to the proper amylopectin structure arrangement promoting its crystallization. DBE can be classified in two classes: isoamylase (ISA) and limit-dextrinase (LDA). ISA1 and ISA2 take part in the amylopectin biosynthesis, whereas ISA3 and LDA function in the starch degradation [10,11,12].

In this regard, to facilitate starch degradation, the external structure of the starch granules must be shifted from a water-insoluble to a soluble state. This is achieved by the action of Glucan, Water Dikinase (GWD) and Phosphoglucan, Water Dikinase (PWD), two plastidial enzymes that mediate these transitions through cycles of phosphorylation of glucans [13,14]. Glucan phosphorylation allows the unfolding of the double helical packaging within the starch, facilitating the access to starch hydrolyzing enzymes [15,16]. GWD catalyzes the phosphorylation of starch at the C-6-OH group of a glucosyl residue, and PWD exerts the same function on C-3-OH. Starch phosphorylation is essential for the mobilization of the transitory starch during the night and is overall regulated by GWD and PWD. In storage starch, in contrast, the role of these two enzymes beyond starch degradation is still unclear. In transgenic potato plants with antisense inhibition of GWD, although a remarkable reduction in the phosphate content was observed, no alterations in the tuber yield or the tuber starch content was detected [17]. Moreover, [18] found that the phosphorylation rate in potato tubers is nearly constant during the whole tuber development, demonstrating at the same time through incubation of tuber discs with Glc and radioisotopes of P^32^ that phosphorylation occurs in de novo synthesis of starch. On the other side, in cassava (*Manihot esculenta*) GWD repression had a significant effect on both leaf and root starch, and the starch excess generated in leaves led to retarded plant and storage root growth [15]. Overall, during starch synthesis phosphorylation also seems to be essential.

Starch dephosphorylation on native starch granules is achieved by the glucan phosphatases SEX4 (Starch Excess 4) and LSF2 (Like Sex Four 2) [19,20]. Arabidopsis plants lacking SEX4 exhibited decreased rates of starch degradation leading to an elevated accumulation of starch in the leaves [21,22].

Other proteins that have been found associated with the starch granules in Arabidopsis and potato tubers are Early Starvation Protein 1 (ESV1) and its homologue Like Early Starvation Protein (LESV) [5,23,24]. Both proteins are involved in the control of starch degradation process by modulating the organization of starch and consequently, affecting the glucan accessibility to catabolic enzymes.

Several proteins, with so far unrevealed catalytic activity, are crucial in the different phases of the starch formation and degradation process. Arabidopsis SS5, for instance, participates in the regulation of the morphology and starch granule number in the chloroplasts, although lacking the glycosyltransferase activity associated with the SSs [25]. Similarly, loss of PTST2 in Arabidopsis causes a phenotype similar to SS4 knockout mutants, in which a reduced number of starch granules is observed [8]. It was proposed that both proteins, PTST2 and SS4, might act together in the starch granule initiation, probably PTST2 providing maltooligosaccharides substrates for SS4 [8].

Plastidial glucan phoshorylase (PHO1), an enzyme with both synthesis and degradation capabilities, has a strong implication on maltooligosaccharide metabolism and, consequently, on the starch metabolic pathway. It catalyzes the addition of glucosyl units to the non-reducing end of α-1,4 linked glucans using glucose-1-phosphate as substrate. In the reverse reaction, PHO1 generates glucose-1-phosphate using α-1,4 linked glucans and orthophosphate as substrates. Transgenic potato tubers with strong repression of PHO1 notably accumulate maltooligosaccharides with a high degree of polymerization, probably due to partial degradation of starch granules during synthesis (e.g., during the trimming of the amylopectin molecules) without being degraded by PHO1 [26]. A distinctive feature of PHO1 is that it has an impact on potato plants during starch biosynthesis when they grow at low temperatures. In potato wild type tubers, increased amounts of short glucans were observed in the chain length distribution of the amylopectin when plants were grown at relatively low conditions (15 °C). Such alteration in the chain length distribution was not observed in transgenic tubers with strong repression of PHO1 [27].

In this work, we investigated the proteins physically interacting with the potato tuber starch under different conditions. First, to analyze if the phosphorylating enzymes GWD and PWD are present during the synthesis of starch, we isolated and characterized the starch proteome in developing tubers, where the synthesis of starch takes place. In addition, as it was demonstrated that PHO1 has a strong implication on the maltodextrin and starch metabolism in potato tubers, the starch proteome of transgenic potato lines repressing the plastidial or cytosolic glucan phosphorylase was characterized and compared with the proteins interacting with the starch of wild type tubers. Moreover, to evaluate the effect of different environmental conditions on the starch proteome, starch-interacting proteins were analyzed in potato tubers stored under different temperatures, revealing that that the expression of specific starch-related proteins are stimulated.

## 2. Results

### 2.1. Proteome Profile of Starch-Interacting Proteins

First, we identified all proteins bound to the starch granules isolated from potato tubers. Proteins interacting with the starch granules were isolated from 500 mg of potato tuber starch, concentrated, and separated by SDS-PAGE. To elucidate the nature of the interaction within the starch granules, protein isolation was conducted in two steps. Weakly interacting proteins to starch (WIP), more likely located on the starch surface, were first removed from the starch through successive washes with SDS. Later, strong interacting proteins (SIP) were recovered by mixing the starch samples with protein extraction buffer while disrupting the starch structure by heating at 99 °C. In Figure 1a, SDS-PAGE separation of SIP is presented. As can be observed, several protein bands in the range of 35 to 250 kDa were detected following Coomassie-blue staining. All the observed bands were excised, trypsin digested, and analyzed by mass spectrometry. For protein identification, peptide masses were searched against the Peptide Mass Fingerprint and MS/MS databases included on the Free Mascot server. From the identified proteins, GBSS is clearly the most abundant starch interacting protein, followed by SS2 (Figure 1a,b). Aside from GBSS, additional starch synthases were found to strongly interact with starch: SS1, SS2, and SS3. Interestingly, LESV, a recently discovered protein implicated in the control of starch degradation process, was found to strongly interact with the starch granule in considerable amounts. Similarly, GWD, PWD, and SEX4, also implicated in starch degradation, were not removable from starch by SDS, but only extracted from the SIP fraction. In addition to mass spectrometry analysis, detection of starch-related proteins using specific antibodies was carried out (all detected proteins are given in Table 1 and Table 2).

Interestingly, one peptide was identified by MS/MS corresponding to the plastidial glucan phosphorylase PHO1 and it was the least abundant of the SIP.

### 2.2. Weak Interacting Proteins to Starch

In Table 2, WIP to starch that were removable using SDS are summarized. Few proteins were recovered by SDS washes and mostly included protease inhibitors. Among these, Multicystatin, Kunitz-type, and Cysteine protease inhibitors were identified. Multicystatin, a 86 kDa endopeptidase with capabilities to inhibit cysteine proteases [28], was consistently the predominant protein band in SDS-PAGE gels of WIP fraction. It has been stated that Multicystatin is well distributed throughout tuber tissue [28,29,30]. Kunitz-type serine protease inhibitor and Cysteine protease inhibitor, both having a mass of nearly 24 kDa, were reported to contain a vacuolar signal peptide [31,32]. In addition, the cytosolic form of glucan phosphorylase, PHO2, was also found to weakly interact with the starch granules. All the WIP removed by SDS had in common that they were not proteins expressed in plastid/amyloplast, and their interaction with starch might have occurred during the starch purification from tubers.

### 2.3. Starch-Interacting Proteins in Transgenic Lines

To evaluate if changes in the profile of the starch-interacting proteins could be observed, we selected two transgenic potato lines for starch isolation, both sustaining a deficiency in the same catalytic activity but in different cell compartments, and we compared it with the starch-interacting proteins in wild-type tubers. Thus, a transgenic line repressing the glucan phosphorylase PHO1 isozyme, which is expressed in the amyloplast/chloroplast, and a second line repressing the cytosolic phosphorylase PHO2 was used. PHO1 is directly involved in starch and maltodextrin metabolism, whereas PHO2 is involved in starch degradation related maltose metabolism in the cytosol [33]. Repression of the targeted enzymes in the transgenic lines was confirmed through Native-PAGE separation of buffer-soluble proteins and subsequent detection of phosphorylase activity (Figure 2).

In Figure 3, starch-interacting proteins extracted from the three different potato lines and separated in SDS-PAGE gel are shown. The protein profile obtained from the transgenic lines and wild type was comparable, with no substantial differences either on the WIP or SIP. However, on the WIP fraction obtained from wild type starch, a clear band with an apparent mass of 100 kDa was observed, which was not visible in the same fraction of the transgenic lines (Figure 3, See WIP fraction). This band was excised and further identified through mass spectrometry as the cytosolic PHO2 (23% sequence coverage; score: 103) and thus, as a contaminating protein. In potato tuber starch with antisense inhibition of PHO2, the corresponding band was not visible. Similarly, in PHO1-repressed lines this additional band was not observed. Despite not having a precise explanation for the contamination of starch with PHO2, it might be related to the surface properties of the isolated starches. In this regard, it was already shown that various mutants related to starch metabolism showed different starch granule surface properties [34].

In addition, we detected starch-bound proteins with antibodies raised against specific enzymes involved in starch degradation (GWD and PWD) or initiation (PTST2) in the three different genotypes. GWD and PWD were not removable from starch by SDS washes and were exclusively found as SIP (Figure 4). PTST2, as was expected for its role on starch initiation, was found in the SIP fraction.

### 2.4. Changes in the Protein Band Pattern of WIP to Starch Was Distinguished When Tubers Were Stored under Different Conditions

As shown, PHO1 was detected as SIP, but not as WIP. However, having as antecedent that PHO1 plays an active role on the starch metabolism in potato and rice plants growing at low temperatures, we assayed the isolation of the starch-interacting proteins after storage of wild type and PHO1-repressed tubers under different temperatures. Right after teir harvest, tubers derived from the same plant were stored at room temperature, warm (37 °C), or cold (4 °C) conditions for 15 days in the dark and starch proteins were later extracted. In the protein fraction corresponding to the SIP, no differences in the protein band patterns were observed. However, as it can be observed in the SDS-PAGE gel image of WIP (Figure 5), disparities on the detection of protein bands were evident comparing the wild type samples against PHO1-repressed samples, and even in the same genotype stored under different conditions. For instance, a nearly 55 kDa protein band (Figure 5, lower arrow) was observed in the PHO1-repressed tuber stored at room temperature and faintly at 37 °C, but not at 4 °C, whereas in wild type samples the same protein band was missing in the three cases. More notably, a protein band having a mass near to 130 kDa (Figure 5 upper arrow) was detected only in the WIP of PHO1-repressed tuber stored at 4 °C. Thus, it seems that the WIP protein pattern was altered among the various conditions, which requires further clarifications.

## 3. Discussion

### 3.1. Proteins Were Mainly Contained in the SIP Fraction

In this work, we explored the starch-interacting proteins in potato starch under different conditions. We classified the starch proteins in two different fractions based on the nature of the interaction considering the way we isolated them from starch. One fraction contained the proteins that were removable from starch by SDS and thus, more likely found loosely interacting with the starch surface, defined in this work as weak interacting proteins (WIP). Strong interacting protein fraction (SIP) was designated to the proteins which were able to recover only after breaking down the granular structure of the starch by heat-induced gelatinization. All identified proteins from the SIP were described as involved in starch metabolism. The protein profile of the SIP observed in the separation gel mainly resembles that obtained by [5]. GBSS was the most profuse protein, followed by SS2. The least abundant SIPs were PWD and PHO1. Interestingly, in the case of GWD and PWD, which are dikinases mainly involved in starch degradation, it would be assumed that they might act on the starch surface based on the catalytic function they realize. However, no traces of either of the phosphotransferases were recovered through consecutive washes of starch samples with SDS. Thereby, both enzymes remained closely interacting within the starch granules.

Finding these phosphorylating dikinases strongly interacting with potato starch is not surprising, as it was previously demonstrated that phosphorylation occurs along the whole storage starch biosynthesis and accumulation in potato tubers [18]. However, the exact role that phosphorylation might play during the synthesis of starch has not been precisely described but is a focus of interest. Aside from their role in starch degradation, probably for additional processes like, for instance, during the arrangement of the starch structure, the activity of these enzymes might be necessary but not totally indispensable, since the tuber yield and tuber starch content in transgenic plants repressing GWD was not compromised [17]. This statement would be especially applicable in this case, where the accumulation of starch mainly takes place in developing tubers. Thus, finding enzymes strictly and only linked to starch degradation would not be expected.

Regarding the WIP proteins, the protein amount that was possible to isolate from starch was considerably lower than that recovered from the SIP fraction (around 8 μg against ~30 μg of SIP), and less protein bands were detected after gel electrophoresis and staining (Figure 5). Of the seven WIP identified in this study, five corresponded to protease inhibitors. It is believed that protease inhibitors in plants are storage proteins that act as defense mechanisms in case of wounding response and represent around 50% of the total soluble proteins in potato tubers [35,36]. Considering this data, the fact that starch proteases were found interacting with starch is not surprising. On the other hand, since these proteins are not expressed in plastids, where starch is accumulated in potato, we assume that the interaction of these protease inhibitors with the starch granules must have occurred during the starch purification procedure from potato tubers rather than naturally occur under physiological conditions. The same assumption can be followed for the cytosolic phosphorylase PHO2, which we also found in the WIP fraction of potato tuber starch. It is well known that PHO2 possesses a strong affinity towards branched glucans, like starch’s amylopectin and glycogen. Therefore, the chance of finding PHO2 interacting with the starch granules after homogenizing of tuber tissue would be very likely.

### 3.2. PHO1 Is Not a Starch Granule-Bound Protein

As we were able to detect only very small amount of PHO1 through MS/MS analysis in SIP but not in WIP, even though PHO1 was found expressed in significant amounts after extraction of total proteins from potato tuber [37], we assayed the detection and location of the plastidial PHO1 in the two protein fractions using an antibody raised specifically against the homologue protein in *A. thaliana.* Proteins were isolated from potato wild-type starch and transgenic PHO1-repressed and PHO2-repressed potato starch samples. A 100 kDa protein band was detected in the starch surface protein fraction (Figure 6a). However, after PMF analysis and data search against the Mascot protein database, a significant match turned out for the cytosolic PHO2, which had a 95 kDa size. The fact that a reduced immunodetection signal was observed in the starch bound protein sample obtained from potato with strong repression of PHO2 was in accordance with the results (Figure 6a). A possible interaction of the anti-*At*PHO1 with *St*PHO2 epitopes could not be excluded since both the plastidial and cytosolic potato phosphorylase protein sequences share highly conserved regions (see Figure 6b). On the other hand, since PHO2 is expressed in the cytosol and possesses strong affinity towards branched glucans like starch’s amylopectin, it can be assumed that its localization within the starch is more related to protein contamination during the starch isolation procedure rather than being a starch-associated protein.

In case of PHO1, failing to detect it within the WIP indicates that, although this enzyme is fundamental in the carbohydrate and starch metabolism, in potato tuber starch it does not remain attached to the granules. This observation is supported by [2], who stated that the plastidial phosphorylase (SP, PHS1, or PHO1) and Disproportionating Enzyme 1 (DPE1), though localized in the plastid stroma, are in general not granule-associated enzymes. This characteristic would be consistent with the function that both enzymes have in modulating and supplying soluble substrates for other enzymes [2].

### 3.3. Effects of Genotypes and Storing Conditions on Proteins Bound to Starch Granules

After identification of the proteins which were regularly found interacting with starch, the question arose whether differences could be traceable on the starch-interacting proteins in potato plants containing alterations in enzymes that are closely linked to the carbohydrate metabolism. To investigate this possibility, we used a transgenic potato line with strong repression of the plastidial glucan phosphorylase activity and another repressing the same activity but in the cytosol. However, no obvious differences were observed after SDS-PAGE separation of the SIP and WIP extracted from PHO1-repressed, PHO2- repressed, and wild-type tubers. In wild-type and the two transgenic potato lines, we also assayed the detection of starch-interacting proteins using specific antibodies. GWD and PWD were found in the SIP fraction in the three different potato starch samples. Protein Targeting to Starch 2 (PTST2), a chloroplastic protein involved in starch granule initiation [8], was also immunodetected in the three different genotypes as SIP having a 43.3 kDa mass.

In a second approach seeking for differences on the starch-interacting proteins, we stored PHO1-repressed and wild-type tubers under different temperature conditions. By these means, different patterns of WIP protein bands were visualized between the samples stored at room temperature, 4 °C, or 37 °C, either in wild-type or PHO1-repressed tubers. Differences were basically observed in the loss/detection of four bands having an apparent mass of ~ 130, ~100, ~50, and ~45 kDa. Unfortunately, due to the low protein concentration of these bands, attempts to identify them were unsuccessful so far. Noteworthy, a nearly 130 kDa protein was detected in the PHO1-repressed protein sample obtained from potato tubers stored at 4 °C, which is close to the molecular weight of PWD. Under such cold storing conditions, the storage starch is more likely under the so called cold-induced sweetening phenomenon, thus the degradation of starch must occur to self-supply reducing sugars. In any case, this band was not detected in wild type tubers stored at 4 °C and a relationship to be attributed to the missing PHO1 activity in the transgenic tuber triggering the expression of PWD is hard to establish.

Considering the findings presented by [23] who reported the presence of ESV1 in both the soluble and insoluble (starch-containing) protein fractions in *A. thaliana* and *N. sylvestris*, and [5], who recently reported the identification of ESV1 (48.9 kDa) in the starch-associated proteins in potato tuber, we consider the probability that one of the WIP bands we observed might correspond to this protein.

Attempts to further investigate the starch proteome under variable environments or simulated stress conditions would be supportive in revealing inducing mechanisms for specific starch-related enzymes and understanding their implication under specific conditions. Like we observed in this assay, expression of different starch-interacting proteins was stimulated by storage of tubers in different temperatures.

## 4. Conclusions

The proteins identified in this study that were strongly interacting with the potato starch granules corresponded to previously described proteins having a specific role in the starch metabolism. Remarkably, GWD and PWD, despite being enzymes commonly associated with the starch degradation, were found to interact strongly with the starch granules in developing tubers. We assume that the activity of GWD and PWD is not exclusive to the starch breakdown but is also required during the synthesis of storage starch. More likely, these two dikinases play a significant role in the regulation of the starch granule architecture by facilitating the cleavage of glucosyl residues by other enzymes. However, the specific role and relevance of these enzymes during starch synthesis must still be investigated.

Furthermore, we demonstrated that the starch proteome profile can be altered during the storage of potato tubers under different temperature conditions. This assay can serve as a milestone to further investigate the starch proteome under different scenarios. This might be helpful, for instance, in the identification of new proteins or revealing which specific starch-related enzymes are preferentially expressed under particular environmental or stress conditions.

## 5. Materials and Methods

### 5.1. Biological Material

#### Potato Plants (*Solanum tuberosum* L.)

Potato wild-type plants (*Solanum tuberosum* L. cv. Desiree), and transgenic lines containing antisense constructs against the plastidial phosphorylase isozymes (PHO1a + b), and the cytosolic phosphorylase isozyme (PHO2) were grown under controlled conditions (12 h light/12 h dark period, 300 μE m^–2^ s^–1^, 22 °C) in a grow chamber. Potato tubers were harvested after 3 months and immediately used for starch isolation, except where otherwise stated.

### 5.2. Antibodies

Polyclonal antibodies used for the detection of AtGWD, AtPWD, and AtPTST2 are described in [8,38,39].

### 5.3. Methods

#### 5.3.1. Starch Isolation

Potato tubers were peeled and homogenized in 100 mL 4 °C cold distilled water per 10–15 g of tuber material using a blender. The mixture was filtered through a 100 µm net sieve and starch was transferred to 50 mL Falcon tubes. Starch was allowed to settle down for 1 h keeping the sample tubes on ice. The supernatant was discarded, and the starch pellet was resuspended in 40 mL cold distilled water, vortexed, and centrifuged at 1500× *g* for 3 min to wash the starch. The supernatant was discarded and the washing procedure was repeated three times. Starch was dried by lyophilization.

#### 5.3.2. Detection of Phosphorylase Activity by Native PAGE

Detection of phosphorylase activity in Native PAGE gels was performed through electrophoretic separation of buffer-soluble proteins extracted from wild type and transgenic potato tubers using the method described by [26].

#### 5.3.3. Extraction of Starch-Interacting Proteins

For protein extraction 500 mg starch was used. Proteins weakly interacting to the starch granule (WIP) were extracted by adding 200 µL 2% (*w*/*v*) SDS per each 100 mg native starch and recovering the supernatant following vortex and centrifugation (20,000× *g* for 10 min). The SDS extraction was repeated twice, and the collected supernatants were pooled in a single tube. Later, starch samples were washed twice with 1 mL distilled water to remove the remaining SDS. Then, proteins which interacted strongly with the starch granules (SIP) were extracted by adding 1 mL of protein extraction buffer (0.2 M Tris, 2% (*w*/*v*) SDS, 20% (*w*/*v*) glycerol, 50 mM DTE, pH 6.8) [5] per sample tube, vortexing, and heating at 99 °C for 20 min. Afterwards, the samples were centrifuged at 20,000× *g* for 10 min and the supernatant was collected.

#### 5.3.4. Concentration of Protein Samples

The two protein fractions containing the WIP and SIP were separately transferred to 10 kDa filter (Merck Millipore^®®^, 5 mL volume capacity) for protein concentration following the supplier manual instructions. Following centrifugation, the volume retained in the filter (nearly 200 μL) was recovered and transferred into a 2 mL Eppendorf tube for acetone precipitation. Protein samples were mixed with four volumes of −20 °C cold acetone, vortexed, and kept in −20 °C for at least 1 h. Then, samples were centrifuged at 20,000× *g* for 5 min. The protein samples were pelleted, and the acetone discarded. Tubes were left uncovered for 10 min for drying.

#### 5.3.5. Determination of Protein Concentration

Starch-interacting protein fractions were resuspended in 100 µL of protein extraction buffer to carry out the Bicinchoninic acid assay (BCA assay; Thermo Fisher Scientific, Waltham, MA, USA) following the supplier protocol. The concentration of protein samples was estimated by comparing with a standard curve generated with known concentrations of Bovine Serum Albumin (BSA). Absorbance was measured at 562 nm as indicated in the supplier manual.

#### 5.3.6. Electrophoretic Separation of Proteins

Starch bound proteins were separated through sodium dodecyl sulfate- polyacrylamide gel electrophoresis (SDS-PAGE) using polyacrylamide concentrations between 7.5 and 10% in the separation gel. The polyacrylamide concentration used in the stacking gel was always 3%. Electrophoresis was performed at 180 V and 40 mA. Following electrophoresis, gels were stained overnight with a commercial Coomassie-blue staining solution (Roti Blue^®®^-Roth, Karlsruhe, Germany).

#### 5.3.7. Quantitation of Protein Bands

The relative intensity of the protein bands detected by Coomassie staining of SDS-PAGE gels was quantified using the image processing program ImageJ 1.51d.

#### 5.3.8. Tryptic Digestion

SDS-PAGE protein bands were prepared for peptide mass fingerprint (PMF) analysis following the protocol described by [40] with some modifications. Bands were excised from the gel, cut into smaller pieces, and distained in 100 µL of 100 mM NH_4_HCO_3_/acetonitrile (1:1) for 15 min with a change of distaining solution in between. Later, the gel pieces were dehydrated in 50 µL acetonitrile for 5 min. The acetonitrile was discarded, and gel pieces were dried in vacuum. The gel pieces were rehydrated with 15 µL of pre-cooled trypsin solution (20–30 ng/µL trypsin sequencing grade reconstituted in 50 mM NH_4_HCO_3_) and incubated on ice for 15 min. Trypsin digestion was performed overnight at 37 °C. For peptide recovering, gel samples were rehydrated with 20 µL distilled water. Later, 20 µL acetonitrile was added and recovered after 5 min. Later, 20 µL of 5% (*v*/*v*) formic acid was added to the gel pieces, incubated for 15 min, and recovered. Finally, a second acetonitrile extraction was performed. Peptide samples were completely dried in a vacuum concentrator. The peptide pellet was resuspended in 10 µL of 20% (*v*/*v*) acetonitrile dissolved in 0.1% (*v*/*v*) trifluoracetic acid (TFA).

#### 5.3.9. Protein Identification

Peptides were analyzed through mass spectrometry using a MALDI-TOF system (Microflex II RFT, Bruker, Bremen, Germany) and MS/MS MALDI-LTQ XL (Thermo Scientific, Waltham, MA, USA). A 0.35 µL peptide sample was spotted on the target plate and covered with the same volume of α-cyano-4-hydroxycinnamic acid matrix (HCCA) prepared as follows: 3.5 mg HCCA dissolved in 1 mL 84% (*v*/*v*) acetonitrile, 13% (*v*/*v*) EtOH, and 3% (*v*/*v*) trifluoroacetic acid 0.1% (*v*/*v*). Peptide mass spectra (*m*/*z* 500–4000 Da) were acquired as positive ions using the reflector mode. The peptide mass values were matched against NCBI and Swissprot databases listed in the MASCOT Peptide Mass Fingerprint and MASCOT MS/MS search platforms (http://matrixscience.com; accessed on 1 June 2023). Trypsin was selected as a digesting enzyme and the peptide tolerance set to 0.6 Da. One or no missed cleavage was allowed for the search and no fixed or variable modifications selected. For MS/MS ion search, the peptide charge was set to +1 and the MS/MS tolerance to 0.6 Da. Results with a MASCOT score above the threshold (*p* < 0.05) were considered significant.

#### 5.3.10. Western Blot

The transfer of proteins from SDS-PAGE gels to nitrocellulose membranes was carried out by placing two Whatman filter papers, the membrane, the protein gel, and two Whatman filter papers in a blotting device (Bio-Rad, Hong Kong, China) following the manual instructions. All components were previously wetted with transfer buffer (25 mM Tris-NaOH pH 8.2, 192 mM glycine, 20% (*v*/*v*) MeOH). Blotting was performed at 12 V for 12 h at room temperature.

#### 5.3.11. Immunodetection of Proteins

Blots were incubated for one hour in blocking TBST solution (100 mM Tris-HCl pH 7.5, 150 mM NaCl, 0.1% (*w*/*v*) Tween 20) containing 3% (*w*/*v*) milk powder and then with primary antibodies (against AtGWD, AtPWD and AtSEX4,) diluted 1:1000 (*v*/*v*) in blocking solution. After one hour, the primary antibody solution was removed and the membranes were washed for five minutes with TBST buffer (100 mM Tris-HCl pH 7.5, 150 mM NaCl, 0.1% (*v*/*v*) Tween 20). This washing step was repeated six times. Later, blot membranes were incubated with secondary antibody (anti-mouse or anti-rabbit conjugated to alkaline phosphatase or horseradish peroxidase (HRP)) for one hour. The membranes were washed again with TBST buffer as mentioned. To visualize alkaline phosphatase activity, BCIP^®®^/NBT (5-bromo-4-chloro-3-indolyl-phosphate/nitro blue tetrazolium) reagent (Sigma-Aldrich) was used following the manual instructions. For HRP conjugated antibodies, signal was detected using the Supersignal West Pico PLUS Chemiluminescent Substrate kit (Thermo Fisher, Waltham, MA, USA) following the manual instructions.

## Figures and Tables

**Figure 1 molecules-28-06219-f001:**
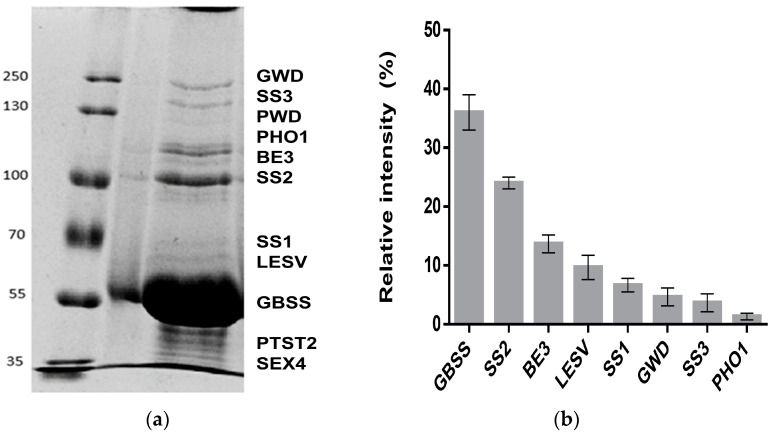
(**a**) A 7.5% SDS-PAGE separation of strong interacting proteins to potato starch. Proteins strongly interacting with starch (SIP) in potato tuber were extracted from 500 mg starch. A volume of 30 μg SIP was separated in a 7.5% SDS-PAGE. For visualization of the protein bands, the separation gel was stained with Roti ^®®^-Blue (Carl Roth). (**b**) Relative intensity of SIP bands in the separation gel was quantified using the image processor ImageJ. Three different gel images of SIP samples extracted from different starch batches were used for quantitation.

**Figure 2 molecules-28-06219-f002:**
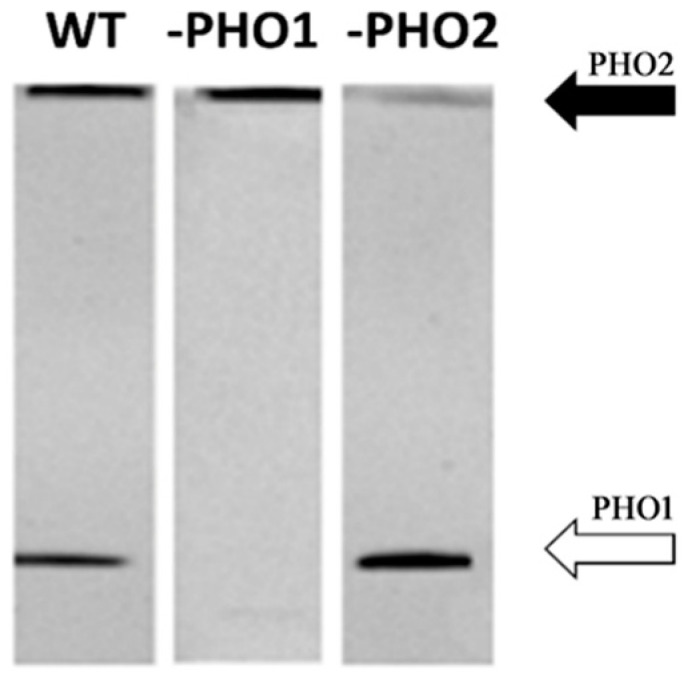
Native-PAGE separation of buffer-soluble proteins extracted from potato (*S. tuberosum*) wild-type tuber, and two transgenic tubers repressing the plastidial (PHO1) and cytosolic (PHO2) phosphorylase enzyme. The procedure used for phosphorylase activity detection was previously described [33]. In each lane, 20 µg buffer-soluble protein from potato tuber was loaded. The separation gel contained 0.2% (*w*/*v*) glycogen. Black arrow indicates the position of cytosolic PHO2; white arrow, plastidial PHO1. In potato tubers with antisense inhibition of PHO1, phosphorylase activity was strongly reduced in the plastidial isoform. In PHO2-repressed tuber, a reduced phosphorylase activity was detected in the cytosolic isoform compared to wild type.

**Figure 3 molecules-28-06219-f003:**
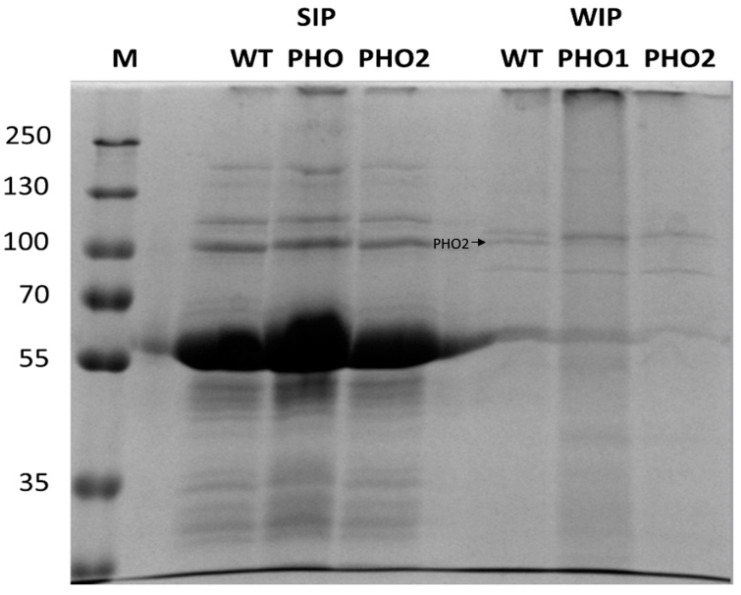
A 9.5% SDS-PAGE of starch-interacting proteins isolated from wild-type, PHO1-repressed and PHO2-repressed transgenic tubers. Starch-interacting proteins isolated from 500 mg starch were separated in a 9.5% SDS-PAGE. Gel was stained with Rotiblue ^®®^ (Carl Roth). A band with an apparent mass of 100 kDa was observed in the protein fraction having a weak interaction with starch (WIP), which corresponded to the cytosolic PHO2.

**Figure 4 molecules-28-06219-f004:**
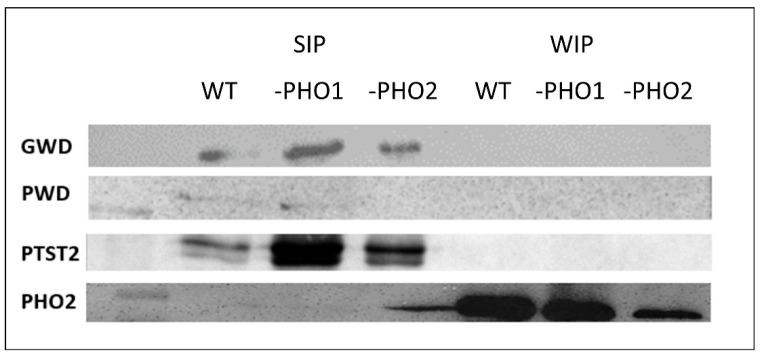
Immunodetection of proteins interacting with starch in potato tuber. Starch-interacting proteins isolated from wild-type, PHO1-repressed, and PHO2-repressed potato tubers were separated in a 9% SDS-PAGE and transferred to a nitrocellulose membrane for western-blot. Specific antibodies raised against *A. thaliana* GWD, PWD, PTST2, and PHO1 were used for immunodetection of starch-interacting proteins in potato. GWD, PWD, and PTST2 were detected as SIP. The cytosolic isoform PHO2 was detected using the AtPHO1 specific antibody.

**Figure 5 molecules-28-06219-f005:**
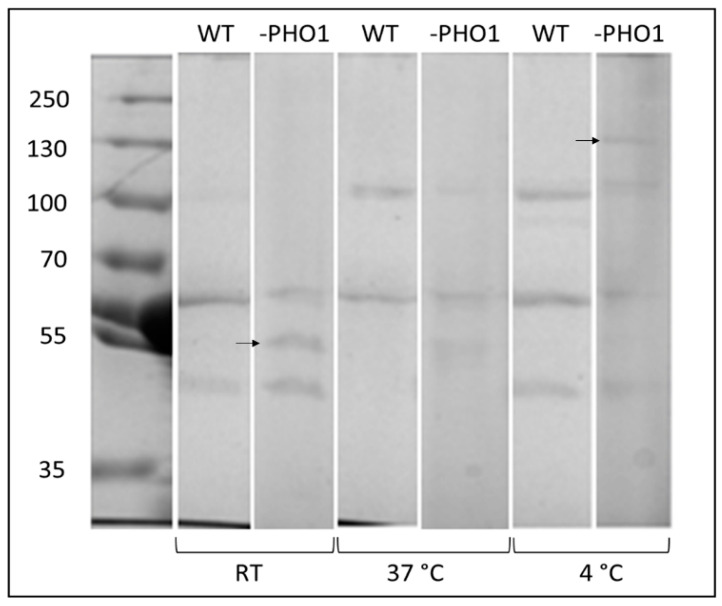
A 7.5% SDS-PAGE separation of weak interacting proteins to starch in wild-type and PHO1-repressed tubers. Immediately after harvesting, wild-type, and PHO1-repressed tubers were stored for 15 days at room temperature (RT), warm (37 °C), or cold (4 °C) conditions. WIP were isolated with 2% (*w*/*v*) SDS and 8.0 µg protein recovered and separated by electrophoresis in a 7.5% polyacrylamide gel.

**Figure 6 molecules-28-06219-f006:**
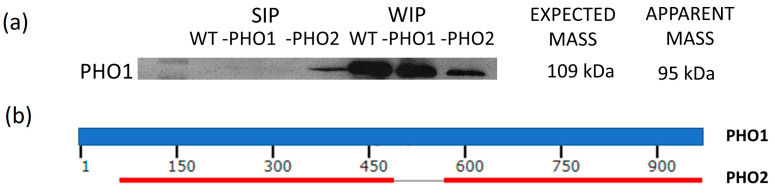
PHO immunodetection in potato starch bound proteins. (**a**) Starch bound proteins separated by SDS-PAGE were transferred to a membrane for immunoblotting and detection using a plastidial phosphorylase (*At*PHO1) specific antibody. A nearly-95-kDa-size protein band was detected in the starch-surface proteins of wild-type and the two transgenic lines with PHO1 and PHO2 antisense inhibition. Through PMF analysis, the corresponding band was identified as the *S. tuberosum* cytosolic phosphorylase isoform PHO2. (**b**) Protein BLAST alignment of *S. tuberosum* PHO1 and PHO2 shows two highly conserved regions among the two sequences. The first region comprises the amino acid 62–487 in PHO1 sequence, and the second region from 566–965.

**Table 1 molecules-28-06219-t001:** Strong interacting proteins to starch (chloro–chloroplast).

Protein Name	Accession Number	Predicted MW (KDa)	Seq. Cov. (%)	Score	Identification Method	Cell Location
GWD	Q9AWA5	163.14	12	105	MALDI-TOF	Chloro
SS3	Q43846	139.02	18	80	MALDI-TOF	Chloro
PWD	D2JRZ6	132.28		N.A.	Immunodetected	Chloro
BE3	P30924	99.02	9	116	MS/MS	Chloro
SS2	Q43847	85.17	27	159	MALDI-TOF	Chloro
SS1	P93568	70.60	13	80	MS/MS	Chloro
GBSS	Q00775	66.58	24	138	MALDI-TOF	Chloro
LESV	Soltu.DM.06G014960.1	63.40	21	77	MS/MS	Chloro
PTST2	XP_006367350	47.26		N.A.	Immunodetected	Chloro
SEX4	Soltu.DM.11G004900.2	41.57		N.A.	Immuno-detected	Chloro

**Table 2 molecules-28-06219-t002:** Weak interacting proteins to starch.

Protein Name	Accession Number	Predicted MW (KDa)	Seq. Cov. (%)	Score	Identification Method	Cell Location
PHO2	P32811	95.05	23	108	MALDI-TOF	Cytosol
LOX 13 Probable linoleate 9S-11 oxygenate	Q43189	96.98	11		MALDI-TOF	Cytosol
Multicystatin	P37842	86.71	16		MALDI-TOF	N.D.
Probable inactive patatin 3-Kuras PT3	Q3YJS9	41.11	8		MS/MS	Vacuole
Kunitz type protease inhibitor	AAB32802.1	24.50		51	MS/MS	Vacuole
Cysteine protease inhibitor 10 (fragment)	O24383	20.96	37		MALDI-TOF	Vacuole
Cysteine protease inhibitor 8 (fragment)	O24384	24.69	31		MALDI-TOF	Vacuole

## Data Availability

Data are available by request from the corresponding author.

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
