# Peer review of "Gradual Analytics of Starch-Interacting Proteins Revealed the Involvement of Starch-Phosphorylating Enzymes during Synthesis of Storage Starch in Potato (Solanum tuberosum L.) Tubers"

_molecules, 2023, doi:10.3390/molecules28176219_

Round 1

Reviewer 1 Report

This study investigated the characterization of the starch interacting proteins in potato (Solanum tuberosum L. cv. Desiree) tubers under variable circumstances. Starch interacting protein profiles of transgenic tubers appeared differently from wild type when tubers were stored under different temperatures, indicating a differential expression in response to changing environmental conditions. I suggest the following minor revisions before publication of this manuscript.

1.       The title is too long, which should concisely be shortened.

2.       Tables are unclear, because they are overlapped with line numbers. The positions of the tables should be corrected.

3. The conclusion section should be created, which can also state some future remarks of this study.

Moderate editing of English language is required.

Author Response

Reviewer 1:

This study investigated the characterization of the starch interacting proteins in potato (Solanum tuberosum L. cv. Desiree) tubers under variable circumstances. Starch interacting protein profiles of transgenic tubers appeared differently from wild type when tubers were stored under different temperatures, indicating a differential expression in response to changing environmental conditions. I suggest the following minor revisions before publication of this manuscript.

  1. The title is too long, which should concisely be shortened.

The title was shortened to:

Gradual analytics of starch interacting proteins revealed the involvement of starch phosphorylating enzymes during synthesis of storage starch in potato (Solanum tuberosum L.) tubers.

  1. Tables are unclear, because they are overlapped with line numbers. The positions of the tables should be corrected.

Tables were adjusted as recommended. In Table 1, Cell location – Chloroplast was abbreviated as Chlorop. to reduce the size of the Table.

  1. The conclusion section should be created, which can also state some future remarks of this study.

Conclusion section was included to the manuscript as precisely recommended.

Reviewer 2 Report

A. Despite high molecular weight, authors have extremely low sequence coverage of SIP proteins in table 1. Numbers like 1% and score 29 for PHO1 are very low and do not merit its inclusion. In addition, sequence coverage numbers are low across the board and strongly limits authors conclusions. Can authors optimize their gel digestion protocol to get better numbers. 

B. Scores are missing for most of the WIP proteins in Table 2.

C. Lines 243-246 = Authors mentioned using mass spectrometry. Can they provide sequence coverage and score as data here. 

D. Lines 243-246 = There are multiple bands at 100 KDa in WT/WIP. Can authors pinpoint on the figure the respective band?

E. Lines 280-281 = This is a negative result. Showing this data might still help readers.

F. Figure 5 = The intensity of bands in this gel is lower in comparison to other figures. Authors do mention the challenges with less amount (3 times). Can authors specify the loading amount in comparison to other figures? How many times authors repeated the procedure and SDS-PAGE?

G. please mark the respective 55 KDa and 130 KDa bands which are discussed in results.

H. Lines 21-25 = Can authors please reframe for more clarity.

I. WIP lane in WT/Room temperature from Figure 5 look different from Figure 3? Can authors explain the reasons? 

The manuscript is well written. Quality of English is good. 

Author Response

Reviewer 2:

  1. Despite high molecular weight, authors have extremely low sequence coverage of SIP proteins in table 1. Numbers like 1% and score 29 for PHO1 are very low and do not merit its inclusion. In addition, sequence coverage numbers are low across the board and strongly limits authors conclusions. Can authors optimize their gel digestion protocol to get better numbers.

A- PHO1 was removed from Table 1 as suggested, since only one peptide was identified that corresponds to PHO1 (described in lines 145-146).

1- Relatively low protein sequence coverage numbers were obtained because the amount of protein bound to starch that can be recovered is very limited. Besides, smearing of GBSS, which is the most abundant protein, during the SDS-PAGE separation was an issue affecting the chances to properly identify peptides related to other proteins during the mass spectrometry analysis, since GBSS peptides were often found in higher ratios. Several approaches and attempts were conducted for the in-gel digestion protocol over an extended period of time.

  1. Lines 243-246 = Authors mentioned using mass spectrometry. Can they provide sequence coverage and score as data here.

The data suggested was added in line 203 of  current version: (23% sequence coverage; score: 103).

  1. Lines 243-246 = There are multiple bands at 100 KDa in WT/WIP. Can authors pinpoint on the figure the respective band?

Figure was adapted pointing out the specific band corresponding to PHO2.

  1. Lines 280-281 = This is a negative result. Showing this data might still help readers.

As we separated the proteins from WT and PHO1 from tubers stored under three different temperatures, in the strong interacting proteins (SIP) we obtained gels with very similar profiles to the observed in Figure 3 for SIP and no additional or missing band can be distinguished. We believed that including these pictures would be redundant and would not contribute with relevant information.

  1. Figure 5 = The intensity of bands in this gel is lower in comparison to other figures. Authors do mention the challenges with less amount (3 times). Can authors specify the loading amount in comparison to other figures? How many times authors repeated the procedure and SDS-PAGE?

Data was added indicating the protein amount loaded (8.0 µg) in the SDS-PAGE of Figure 5. This experiment was performed only once.

  1. please mark the respective 55 KDa and 130 KDa bands which are discussed in results.

Figure was adapted with arrows indicating the bands. Also, in the text was specified as follows:

For instance, a nearly 55 kDa protein band (Figure 5, lower arrow) was observed in the PHO1-repressed tuber stored at room temperature and faintly at 37 °C, but not at 4 °C. Whereas, in wild type samples the same protein band was missing in the three cases. More notably, a protein band having a mass near to 130 kDa (Figure 5 upper arrow) was detected only in the WIP of PHO1-repressed tuber stored at 4 °C.

  1. Lines 21-25 = Can authors please reframe for more clarity.

The lines were adapted as follows:

Most of the proteins strongly interacting to the potato starch granules corresponded to proteins involved in the starch metabolism. GWD and PWD, two dikinases associated with starch degradation, were consistently found bound to the starch granules. This indicates that their activity is not only restricted to degradation but is also essential during storage starch synthesis.

  1. WIP lane in WT/Room temperature from Figure 5 look different from Figure 3? Can authors explain the reasons?

First, in Figure 3 the WIP were separated in a 9.5 % SDS PAGE, while in Figure 5 was 7.5%. Second, the potato tubers used for the data presented in Figure 3 were harvested and immediately used for starch and protein extraction. On the other hand, in Figure 5 the tuber was stored in the dark during 15 days at room temperature before starch/protein extraction.

Round 2

Reviewer 2 Report

Authors have satisfactorily answered my comments.